



# Global modeling of cloudwater acidity, rainwater acidity, and acid inputs to ecosystems

Viral Shah[1], Daniel J. Jacob[1,2], Jonathan M. Moch[2], Xuan Wang[1,a], Shixian Zhai[1]

[1]Harvard John A. Paulson School of Engineering and Applied Sciences, Harvard University, Cambridge, MA, USA.
[2]Department of Earth and Planetary Sciences, Harvard University, Cambridge, MA, USA.
[a]Now at School of Energy and Environment, City University of Hong Kong, Hong Kong SAR, China.

*Correspondence to*: Viral Shah (vshah@seas.harvard.edu)

**Abstract.** Cloudwater acidity affects the atmospheric chemistry of sulfate and organic aerosol formation, halogen radical cycling, and trace metal speciation. Rainwater acidity including post-depositional inputs adversely affects soil and freshwater

ecosystems. Here we use the GEOS-Chem model of atmospheric chemistry to simulate the global distributions of cloud- and rainwater acidity, and the total acid inputs to ecosystems from wet deposition. The model accounts for strong acids ($H_2SO_4$, $HNO_3$, $HCl$), weak acids ($HCOOH$, $CH_3COOH$, $CO_2$, $SO_2$), and weak bases ($NH_3$, dust and sea salt aerosol alkalinity). We compile a global dataset of cloudwater pH measurements for comparison with the model. The global mean observed cloudwater pH is $5.2 \pm 0.9$, compared to $5.0 \pm 0.8$ in the model, with a range of 3 to 8 depending on region. The lowest values

are over East Asia and the highest values are over deserts. Cloudwater pH over East Asia is low because of large acid inputs ($H_2SO_4$, $HNO_3$), despite $NH_3$ and dust neutralizing 70% of these inputs. Cloudwater pH is typically 4–5 over the US and Europe. Carboxylic acids account for less than 25% of cloudwater $H^+$ in the northern hemisphere on an annual basis, but 25–50% in the southern hemisphere and over 50% in the southern tropical continents where they push the cloudwater pH below 4.5. Anthropogenic emissions of $SO_2$ and $NO_x$ (precursors of $H_2SO_4$ and $HNO_3$) are decreasing at northern mid-latitudes, but

the effect on cloudwater pH is strongly buffered by $NH_4^+$ and carboxylic acids. The global mean rainwater pH is 5.5 in GEOS-Chem, higher than the cloudwater pH because of dilution and below-cloud scavenging of $NH_3$ and dust. GEOS-Chem successfully reproduces the rainwater pH observations in North America, Europe, and eastern Asia. Carboxylic acids, which are undetected in routine observations due to biodegradation, lower the annual mean rainwater pH in these areas by 0.2 units. The acid wet deposition flux to terrestrial ecosystems taking into account the acidifying potential of $NO_3^-$ and $NH_4^+$ in N-

saturated ecosystems exceeds 50 meq $m^{-2}$ $a^{-1}$ in East Asia and the Americas, which would affect sensitive ecosystems. $NH_4^+$ is the dominant acidifying species in wet deposition, contributing 41% of the global acid flux to continents under N-saturated conditions.

## 1 Introduction

Cloudwater acidity ($H^+$ concentration) affects global atmospheric chemistry in a number of ways. It controls the rates of
30 aqueous-phase reactions that (1) oxidize sulfur dioxide ($SO_2$) to sulfate aerosols (Martin et al., 1981; Calvert et al., 1985), (2)



oxidize dissolved organic compounds to less volatile forms leading to secondary organic aerosols (Ervens et al., 2011; Herrmann et al., 2015), and (3) convert halides into halogen radicals (von Glasow and Crutzen, 2003; Platt and Hönninger, 2003). It affects the solubility and bioavailability of iron in aerosol particles and thus the input of this micronutrient to marine ecosystems (Mahowald et al., 2005). Acidic deposition has a range of environmental effects on soil and freshwater ecosystems (Driscoll et al., 2001). Cloud- and rainwater acidity is affected in a complex way by natural and anthropogenic emissions, but there has been little effort so far to evaluate the ability of global models to represent this. Here we present such an evaluation with the GEOS-Chem atmospheric chemistry model and go on to discuss the factors controlling cloud- and rainwater acidity on a global scale.

Cloud- and rainwater $H^+$ concentrations are determined by the balance between dissolved acids ($H^+$ donors) and bases ($H^+$ acceptors). Sulfuric acid ($H_2SO_4$), nitric acid ($HNO_3$), and hydrogen chloride (HCl) are the major strong acids in the atmosphere, and they dissociate completely in cloud- and rainwater. The major weak acids are $CO_2$, $SO_2$, and carboxylic acids including formic acid (HCOOH), and acetic acid ($CH_3COOH$). Ammonia ($NH_3$) and alkaline dust particles are the major bases. Atmospheric acidity is commonly referenced to the $CO_2$–$H_2O$ system (pH 5.6 at current $CO_2$ levels), with lower pH referred to as acidic conditions and higher pH as alkaline conditions. Cloudwater pH generally varies between 3 and 7, with highly acidic cloudwater typically found in industrialized areas with high $SO_2$ and nitrogen oxides ($NO_x$) emissions, and alkaline cloudwater found in agricultural and dusty areas (Warneck, 2000; Pye et al., 2020). Rainwater pH varies in a similar pattern (Dentener and Crutzen, 1994; Vet et al., 2014) but differs from cloudwater pH because of dilution (Weathers et al., 1988; Bormann et al., 1989), riming (Collett et al., 1993), below-cloud scavenging (Castillo et al., 1983; Zinder et al., 1988; Ayers and Gillett, 1988), and oxidation chemistry within raindrops (Overton et al., 1979; Graedel and Goldberg, 1983).

The chemical and physical processes governing cloud- and rainwater acidity have been well-established since the 1980s (Morgan, 1982; NRC, 1983; Stumm et al., 1987). They have been incorporated in many regional models focused on acid deposition (Chang et al., 1987; Venkatram et al., 1988; Carmichael et al., 1991; Hass et al., 1993; Olendrzynski et al., 2000; Langner et al., 2005) and global models focused on sulfur and nitrogen deposition (Dentener and Crutzen, 1994; Rodhe et al., 1995; Bouwman et al., 2002; Rodhe et al., 2002; Tost et al., 2007; Paulot et al., 2018). A few global modeling studies have focused on rainwater pH (Dentener and Crutzen, 1994; Rodhe et al., 1995, 2002; Tost et al., 2007). These models calculated rainwater $[H^+]$ from the rainwater concentrations of $SO_4^{2-}$, $NO_3^-$, $NH_4^+$, $HCO_3^-$, and $CO_3^{2-}$ using ionic charge balance. Rodhe et al. (2002) also included dust alkalinity. None included carboxylic acids, which are known to be important but biodegrade rapidly after deposition (Keene et al., 1983; Keene and Galloway, 1984).

Cloudwater pH has received less attention in models. Some current global atmospheric chemistry models assume a constant cloudwater pH for aqueous reactions (S. Watanabe et al., 2011; Søvde et al., 2012), while others calculate it explicitly from the balance of acids and bases but again generally neglecting dust alkalinity and carboxylic acids (Tost et al., 2007; Huijnen



et al., 2010; Myriokefalitakis et al., 2011; Alexander et al., 2012; Lamarque et al., 2012; Simpson et al., 2012). Pye et al. (2020) presented the cloudwater pH values simulated by five such models and included a limited comparison with observations. They found large differences among models particularly in dusty areas where pH estimates varied by 3–4 units. All models showed large systematic biases compared to observations. In light of these findings, Pye et al. (2020) highlighted the need for improvements in the cloudwater simulations including further evaluation with observations.

Here we present a global analysis of cloud- and rainwater pH in the GEOS-Chem model with an improved cloudwater pH calculation, including in particular carboxylic acids and dust alkalinity, and an explicit rainwater pH calculation. We evaluate the simulation with extensive cloud- and rainwater measurements and determine the sources of acidity and alkalinity in different parts of the world. We examine the buffering effects of $NH_3$ and carboxylic acids on cloudwater pH, and the changes

in acid inputs to terrestrial ecosystems from post-depositional processes.

## 2 Model description

We use the GEOS-Chem atmospheric chemistry model (www.geos-chem.org) version v11-02 with a number of modifications, some from more recent GEOS-Chem versions and some specifically from this work. The model is driven by assimilated meteorological fields from the NASA Global Modeling and Assimilation Office's Modern-Era Retrospective analysis for

Research and Applications, version 2 (MERRA-2) system (Gelaro et al., 2017). These fields include in particular cloudwater liquid and ice content, cloud volume fraction, and 3-D liquid and ice precipitation fluxes, updated every three hours. GEOS-Chem includes detailed $NO_x$-hydrocarbon-aerosol-halogen chemistry (Mao et al., 2013; Kim et al., 2015; Travis et al., 2016; Sherwen et al., 2016), and here we have added recent halogen updates (X. Wang et al., 2019). The model distinguishes between fine and coarse aerosol but does not otherwise include aerosol microphysics. Wet deposition follows the algorithm of Liu et

al. (2001) including rainout (in-cloud scavenging), washout (below-cloud scavenging), and scavenging in convective updrafts, with updates by Q. Wang et al. (2011) and Amos et al. (2012). Dry deposition follows a standard resistance-in-series scheme (Wesely, 1989; Y. Wang et al., 1998). We conduct the simulation on a global 4º latitude × 5º longitude grid for the year 2013 following an initialization period of one year.

### 2.1 Emissions and acid-producing chemistry

Here we describe the GEOS-Chem emissions and chemistry most relevant to the simulation of cloud- and rainwater acidity. Emissions are calculated by the Harvard-NASA Emissions Component (HEMCO) (Keller et al., 2014). Default anthropogenic emissions of $SO_2$, $NO_x$, and $NH_3$ are from the global CEDS emissions inventory for 2013 (Hoesly et al., 2018). They are superseded by regional emissions inventories including MIX over Asia for 2010 (M. Li et al., 2017), MEIC over China for 2013 (Zheng et al., 2018), NEI 2011 over the US scaled to 2013 (Travis et al., 2016; U.S. Environmental Protection Agency,

2018), APEI over Canada for 2013 (van Donkelaar et al., 2008), EMEP 2008 over Europe scaled to 2013 (EEA, 2019) and





DICE over Africa for 2013 (Marais and Wiedinmyer, 2016). Ship $SO_2$ emissions are from Eyring et al. (2005). Ship $NO_x$ emissions are from the ICOADS inventory (C. Wang et al., 2008) and are pre-processed with the PARANOX ship plume model (Vinken et al., 2011; Holmes et al., 2014). Aircraft emissions are from the AEIC inventory (Stettler et al., 2011). Biomass burning emissions are from GFED v4 (van der Werf et al., 2017). Natural emissions include $NO_x$ from lightning (L. Murray et al., 2012) and soil (Hudman et al., 2012), volcanic $SO_2$ (Fisher et al., 2011), marine dimethyl sulfide (DMS) (Breider et al., 2017), and $NH_3$ from oceans, natural soils, and human population (Bouwman et al., 1997). Sea salt aerosol emissions in two size classes (fine and coarse) follow Jaeglé et al. (2011). Dust emissions include desert and semi-desert sources (Fairlie et al., 2007; Ridley et al., 2013), and combustion and industrial sources (Philip et al., 2017) in four size classes (one fine and three coarse). Biogenic volatile organic compounds (VOC) emissions are from MEGAN (Guenther et al., 2012; Hu et al., 2015).

Sulfur chemistry in GEOS-Chem includes oxidation of DMS to $SO_2$ and methanesulfonic acid (MSA), gas-phase oxidation of $SO_2$ to $H_2SO_4$, and aqueous-phase oxidation of $SO_2$ to $H_2SO_4$ in clouds, rain, and alkaline sea salt aerosols (Alexander et al., 2005, 2009; Q. Chen et al., 2017). Nitrogen chemistry includes oxidation of $NO_x$ to $HNO_3$ in the gas phase, and in the aqueous phase of aerosols and clouds (McDuffie et al., 2018; Holmes et al., 2019). Tropospheric HCl is mainly from acid displacement reactions on sea salt aerosols (X. Wang et al., 2019).

$HNO_3$, HCl, and $NH_3$ are semi-volatile and their gas-particle partitioning affects their scavenging efficiency in cloud- and rainwater (Amos et al., 2012). We calculate this partitioning at bulk thermodynamic equilibrium using ISORROPIA II for the $H_2SO_4$-$HNO_3$-HCl-$NH_3$-NVC metastable aqueous system, where NVC represents the non-volatile cations from fine-mode sea salt aerosol (X. Wang et al., 2019). The uptake of $HNO_3$ and release of HCl (acid displacement) on coarse-mode sea salt aerosol is treated as a kinetic process (X. Wang et al., 2019).

## 2.2 Simulation of HCOOH and $CH_3COOH$

The most important carboxylic acids for cloud- and rainwater acidity are HCOOH ($pK_a = 3.8$ at 298K) and $CH_3COOH$ ($pK_a = 4.8$ at 298K) (Morgan, 1982; Keene et al., 1983). They are present in the atmosphere at comparable concentrations (Talbot et al., 1997) but HCOOH is more important for contributing to acidity because of its higher Henry's law solubility and lower $pK_a$. Sources of these acids include secondary production from VOC oxidation and direct emissions from biomass burning, fossil-fuels, soils, and vegetation (Khare et al., 1999), but these are poorly understood and models greatly underestimate atmospheric concentrations (Paulot et al., 2011; Stavrakou et al., 2012; Millet et al., 2015; Khan et al., 2018). Here we use the previous GEOS-Chem HCOOH simulation by Millet et al. (2015) which scales up the biogenic emissions from the MEGAN inventory (Guenther et al., 2012) in order to fit atmospheric observations over the US. This yields a global HCOOH source of 1900 Gmol $a^{-1}$. Stavrakou et al. (2012) previously estimated a global HCOOH source of 2200–2600 Gmol $a^{-1}$ from inversion of satellite data. In addition, we assume a minimum background mixing ratio of 100 pptv (50 pptv south of 60°S), based on


measurements in the marine boundary layer and the free troposphere (Arlander et al., 1990; Talbot et al., 1990, 1997; Legrand et al., 2004) and satellite-derived free troposphere HCOOH columns over marine areas of $1-2\times10^{15}$ molecules cm$^{-2}$ (Franco et al., 2020).

Our CH$_3$COOH simulation follows the standard GEOS-Chem mechanism (Mao et al., 2013; Travis et al., 2016) without further improvement, except that the minimum background CH$_3$COOH concentration is also taken to be 100 pptv (50 pptv south of

35 60°S), based on observations in the marine boundary layer and the free troposphere (Arlander et al., 1990; Talbot et al., 1990, 1997; Helas et al., 1992; Franco et al., 2020). The global simulated CH$_3$COOH source is 1000 Gmol a$^{-1}$. Other modeling studies attempting to fit CH$_3$COOH observations have estimated a source in the range 1700–3900 Gmol a$^{-1}$ (Baboukas et al., 2000; Khan et al., 2018).

Figure 1 compares annual mean GEOS-Chem wet deposition fluxes of HCOOH and CH$_3$COOH with observations from the compilations of Vet et al. (2014) and Keene et al. (2015). We find that the mean GEOS-Chem HCOOH flux (7.5 mmol m$^{-2}$ a$^{-1}$) is consistent with the mean of the observations (6.9 mmol m$^{-2}$ a$^{-1}$). The model captures the high fluxes observed in the tropical continents where there are large biogenic sources, and the low fluxes observed at marine sites. GEOS-Chem underestimates the CH$_3$COOH flux by a factor of 4. The observed patterns of HCOOH and CH$_3$COOH fluxes are similar,

suggesting that model CH$_3$COOH could be corrected similarly to HCOOH in future work by scaling up biogenic emission.

### 2.3 Calculation of cloud- and rainwater composition and pH

Cloudwater composition is computed locally in each grid cell containing liquid cloudwater over 30-min time steps using the in-cloud liquid water content and cloud volume fraction from MERRA-2. Dissolution of gases in the cloud droplets follows the Henry's law constants of Table 1 and acid/base dissociation constants of Table 2. We assume that 70% of fine aerosol mass

and 100% of coarse aerosol mass are partitioned into cloudwater (Hegg et al., 1984; Alexander et al., 2012). Sulfate-nitrate-ammonium and sea salt particles dissolve completely in cloudwater, and the alkaline component of the dust particles also dissolves. Freshly emitted sea salt particles contain an alkalinity of 0.07 eq kg$^{-1}$ (Alexander et al., 2005), while freshly emitted dust particles contain an alkalinity of 4.5 eq kg$^{-1}$ based on the assumption of 7.1% Ca$^{2+}$ and 1.1% Mg$^{2+}$ by dry mass (Engelbrecht et al., 2016) with CO$_3^{2-}$ as anion. Sea salt NVCs are expressed as Na$^+$ equivalents, while dust NVCs are expressed

as Ca$^{2+}$ equivalents. The upper limit of Ca$^{2+}$ concentration is set by formation of CaCO$_3$(s).

The calculation of cloudwater composition in the cloudy fraction of each grid cell assumes a closed system where the summed concentrations of gas and cloudwater species in Table 3 are conserved, and the partitioning is then computed following the equilibria of Tables 1 and 2. The calculation is done by solving the electroneutrality equation iteratively using Newton's

method (Moch et al., 2020). This improves on the original calculation of cloudwater composition in GEOS-Chem (Alexander



et al., 2012) through the inclusion of additional acidic and alkaline species (HCl, HCOOH, CH₃COOH, NVCs) and using a more stable numerical solver.

We will present results as time averages (mainly annual) and spatial averages (vertical or zonal). The time- and space- averaged
cloudwater [$H^+$] cannot be calculated directly from the [$H^+$] computed at each model time step in each grid cell because [$H^+$] is a non-conservative quantity controlled by the other acidic and basic species in cloudwater (Liljestrand, 1985). Therefore, we calculate the average cloudwater [$H^+$] from the corresponding volume weighted average (VWA) concentrations of the cloudwater ions. We assume that all acids and bases except carbonates are conserved in the aqueous phase. For HCOOH and CH₃COOH, the total (dissociated + undissociated) amounts are assumed to be conserved. Thus, the time- and space-averaged
cloudwater [$H^+$] is given by:

$$\overline{[H^+]} = 2\overline{[SO_4^{2-}]} + \overline{[NO_3^-]} + \overline{[Cl^-]} + \overline{[HSO_3^-]} + 2\overline{[SO_3^{2-}]} + \overline{[HCOO^-]} + \overline{[CH_3COO^-]} + \overline{[HCO_3^-]} - \overline{[NH_4^+]} - 2\overline{[Ca^{2+}]} - \overline{[Na^+]} \quad (1)$$

where $\overline{[A]}$ represents the VWA molar concentration in cloudwater of species A over the time period and spatial domain of interest. We calculate $\overline{[A]}$ from the concentration of the species, $[A]_{i,j}$, and the cloud liquid water content, $L_{i,j}$, at each model time step $i$ and grid cell $j$:

$$\overline{[A]} = \frac{\sum_{j=1}^m \sum_{i=1}^t L_{i,j}\,[A]_{i,j}}{\sum_{j=1}^m \sum_{i=1}^t L_{i,j}} \quad (2)$$

where $[1, t]$ is the averaging time period and $[1, m]$ is the ensemble of grid cells included in the average. [$CO_3^{2-}$] and [$OH^-$] are negligible compared to other anions over the range of cloudwater pH values (< 8). $\overline{[HCOO^-]}$ is calculated from the total aqueous concentration, $\overline{[HCOOH]}_{aq,T} = \overline{[HCOOH]}_{aq} + \overline{[HCOO^-]}$, as follows:

$$\overline{[HCOO^-]} = \left(\frac{K_a}{K_a + \overline{[H^+]}}\right) \overline{[HCOOH]}_{aq,T} \quad (3)$$

where $K_a$ is the HCOOH(aq)/HCOO⁻ acid dissociation constant from Table 2 computed at the average cloudwater temperature for the time period and spatial domain. The same procedure is used for $\overline{[CH_3COO^-]}$. $\overline{[HCO_3^-]}$ is calculated from equilibrium with atmospheric CO₂ as follows:

$$\overline{[HCO_3^-]} = \frac{H_{CO_2} K_{c1} P_{CO_2}}{\overline{[H^+]}} \quad (4)$$

where $H_{CO_2}$ and $K_{c1}$ are the Henry's law coefficient for CO₂ and the CO₂(aq)/HCO₃⁻ acid dissociation constant, respectively,
at the average cloudwater temperature for the period and domain (Tables 1 and 2). $P_{CO_2}$ is the CO₂ partial pressure, taken to be 390 ppm as representative of 2013. Since $\overline{[HCOO^-]}$, $\overline{[CH_3COO^-]}$, and $\overline{[HCO_3^-]}$ calculated in this way depend on $\overline{[H^+]}$, Eq. (1) is cubic in $\overline{[H^+]}$. The time- and space-averaged pH ($\overline{pH}$) is calculated from $\overline{[H^+]}$:

$$\overline{pH} = -\log_{10}\left(\overline{[H^+]}\right) \quad (5)$$





There is some arbitrariness in assuming that $NH_3$, $SO_2$, and carboxylic acids do not equilibrate with the gas phase during averaging. We examined the sensitivity to this assumption by assuming alternatively that $NH_{3T}$, $SO_{2T}$, $HCOOH_T$ and $CH_3COOH_T$ as defined in Table 3 (sum of gas-phase and aqueous-phase concentrations) are conserved and recalculating the gas–cloudwater equilibrium for the time-averaged conditions. We find no significant difference in the computed $\overline{[H^+]}$.

Calculation of rainwater VWA composition including $\overline{[H^+]}$ follows the same approach as for cloudwater. In that case we use the model-archived wet deposition fluxes including contributions from in-cloud and below-cloud scavenging. We assume that $SO_2$ is instantly oxidized by $H_2O_2$ (as available) in rainwater and is scavenged as $SO_4^{2-}$. As with cloudwater, the maximum $[Ca^{2+}]$ is set by the formation of $CaCO_3(s)$. Rainwater pH measurements are generally reported as monthly means and do not account for $HCOOH$ and $CH_3COOH$, which biodegrade rapidly. To compare with measurements, we calculate a monthly $\overline{[H^+]}$

00   for each grid cell by removing $\overline{[HCOO^-]}$ and $\overline{[CH_3COO^-]}$ from the charge balance in Eq. (1). From there we calculate the annual precipitation VWA pH:

$$\overline{pH_{VW}} = -\log_{10}\left(\frac{\sum_{k=1}^{12} P_k \overline{[H^+]}_k}{\sum_{k=1}^{12} P_k}\right) \tag{6}$$

where $\overline{[H^+]}_k$ is the mean rainwater $[H^+]$ and $P_k$ is the precipitation depth for month $k$.

## 3 Results and discussion

05   ### 3.1 Global distribution of cloudwater pH and composition

Figure 2 shows the global distribution of cloudwater pH as simulated by GEOS-Chem and as measured at mountain sites, coastal sites (marine fog), and from aircraft campaigns (Table A1). We exclude continental fogwater measurements because of their sensitivity to local emissions. The measurements span the period from 1980 to 2018, but near source regions we only include observations made after 2006. Some of the observations are taken from the Pye et al. (2020) compilation.

The top panel shows the mean observed and simulated pH values. The observed values are as reported in the literature, where the computation of the mean is generally based on VWA $[H^+]$. The model values are annual means below 700 hPa ($\approx$3 km above sea level) for the year 2013. The bottom panel shows the observations aggregated by regions and arranged in ascending pH order, along with the corresponding GEOS-Chem values. For comparison with the observations, the model is sampled at

the locations and months of the measurements.

The global mean cloudwater pH in the observations is 5.2 ± 0.9, compared to 5.0 ± 0.8 in the model. Annual mean values in Fig. 2 range from 3 to 8, showing distinct spatial patterns. The spatial patterns can be understood from the distributions of acidic and basic ions shown in Fig. 3. $Na^+$ and $Cl^-$ are major constituents of cloudwater in marine and coastal areas but are not





shown in the Fig. 3 because they are largely in balance and have little net effect on pH. Figure 3 also shows the percent contribution of each ion to the total anion or cation concentrations (contour lines).

The lowest pH values are generally over East Asia, both in the observations and in the model, with values mostly below 4.3 in eastern and southern China and Japan. This is due in GEOS-Chem to extremely high $[SO_4^{2-}]$ and $[NO_3^-]$, and despite 70% of

these anions being balanced by $NH_4^+$ and dust cations. pH values remain low over the North Pacific (~4.5), despite much lower acid inputs, because less than half of the acidic anions are balanced by bases.

Over the US and Europe, average pH values are in the range 4–5 both in the observations and in the model, with most of the acidity as $NO_3^-$, and $NH_4^+$ balancing over 50% of the acidic anions. The model shows higher values in the central US (~5.5)

because of $NH_3$ and dust emissions. Summertime observations in the southwest US by Hutchings et al. (2009) show a pH of 6.3 due to large dust influence, but this seasonal dust emission is underestimated in GEOS-Chem (Fairlie et al., 2007). Over southern Europe, about 25% of the base cations are from Saharan dust. $NH_4^+$ is the main cation elsewhere in Europe. $SO_4^{2-}$ is the dominant acidic component over the northern midlatitude oceans because of the oceanic source of $SO_4^{2-}$ from the oxidation of DMS and because continental influence extends further for $SO_4^{2-}$ than for $NO_3^-$ (Heald et al., 2006). Over the Arctic, the

simulated pH is much lower (4–4.5) because of long-range transport of acidic species and less than 50% neutralization (Fisher et al., 2011).

The carboxylate ions $HCOO^-$ and $CH_3COO^-$ account for less than 25% of $H^+$ throughout the extratropical northern hemisphere (Fig. 3). The carboxylic acids are more important relative contributors to $H^+$ in the tropics and in the southern hemisphere, 

exceeding 50% in some areas of the tropical continents and southern midlatitudes. Ayers and Gillett (1988) found that carboxylic acids were responsible for observed cloudwater pH below 4 over tropical Australia, but GEOS-Chem underestimates carboxylic acids in that region (Fig. 1). Carboxylates were not measured in the Ecuador cloudwater measurements (Makowski Giannoni et al., 2016), but we find from the model that the carboxylic acids contribute about 50% of the $H^+$. GEOS-Chem shows similar pH values as observed at Cape Grim (mean of 5.5), reflecting the low concentrations of

acidic and basic species from continental sources. Cloudwater pH sampled on the Antarctic coast also has a mean pH of 5.5 (Saxena and Lin, 1990) but the model is much lower over the Antarctic coast because of $SO_4^{2-}$ from oxidation of DMS. This may be because of sea salt alkalinity from blowing snow that is not accounted for in the model (Huang et al., 2018).

Alkaline cloudwater (pH > 5.6) is found in the observations over western India, Tibet, and Morocco, consistent with the model

where the alkalinity is mainly from dust. GEOS-Chem simulates pH of 6–8 over the area extending from the Sahara to the Gobi Desert. The transport of dust alkalinity from North Africa raises the cloudwater pH in the Caribbean to above 5.5, both in the observations (Gioda et al., 2011) and in the model.





Figure 4 shows the zonal mean distributions of cloudwater pH and cloud liquid water content (contour lines). In addition to the latitudinal variations described previously, cloudwater pH increases as the cloud liquid water content increases because of the effect of dilution. Liquid water content peaks at about 900 hPa, and decreases at higher altitudes, and this largely explains the mean variation of pH with altitude.

Pye et al. (2020) showed in their review the annual mean tropospheric-column cloudwater pH from three models: CMAQ (northern hemisphere only), TM4-ECPL, and GEOS-Chem. They calculated the annual mean pH using VWA [$H^+$] rather than Eq. (1). We find that their calculation method underestimates the pH over alkaline regions by 1–3 units but has little error for acidic regions. The GEOS-Chem results shown in Pye et al. (2020) follow the previous cloudwater [$H^+$] calculation from Alexander et al. (2012), which is different from our calculation (Sect. 2). The resulting pH values are comparable over industrialized regions but differ by up to 2 units over tropical forests and deserts because carboxylic acids and dust alkalinity

were not previously included. pH values were generally higher than 6 over the oceans in the previous GEOS-Chem simulation because of an error in the numerical solver. The pH values shown by Pye et al. (2020) for TM4-ECPL are lower than our simulation by 1–2 units over most areas, and would be inconsistent with the observations shown in Fig. 2. Our pH estimates are closest to those from CMAQ, which was the only model in Pye et al. (2020) that included dust alkalinity and HCOOH in the cloud water pH calculation. The zonal mean cloud water pH values from the ECHAM5/MESSy1 model presented by Tost

et al. (2007) are 0.2–0.3 units higher than our values, likely because of their omission of carboxylic acids.

   Emissions of $SO_2$ and $NO_x$ in the northern mid-latitude continents are decreasing rapidly because of air quality concerns (Hoesly et al., 2018; Zheng et al., 2018). Considering that $NH_3$ and dust neutralize presently balance over 50% of the acidic anions over these continents (Fig. 3), one might expect large increases in cloudwater pH as $SO_2$ and $NO_x$ emissions decrease.

There is however a large buffering effect from the semi-volatile carboxylic acids and $NH_3$. Consider the case of the US. Figure 5 shows the mean simulated cloudwater composition for 2013 over the continental US, and the change in composition resulting from a factor of two decrease in strong acidity ($SO_4^{2-}$, $NO_3^-$). The total concentrations of carboxylic acids ($RCOOH_T$ = $HCOOH_T + CH_3COOH_T$) and ammonia ($NH_{3T}$) are held at 2013 levels and the gas–cloudwater equilibrium is recalculated. For 2013, the mean cloudwater pH is 4.7, a level at which only one-fourth of $RCOOH_T$ is present as carboxylate ions, and most of

the $NH_{3T}$ is present as $NH_4^+$. Decreasing the strong acidity by half triples the dissociated fraction of $RCOOH_T$ and volatilizes a significant fraction of $NH_4^+$, which limits the increase in cloudwater pH to 5.7. Without this buffering effect, the pH would have increased by 2.1 units to 6.8. $NH_4^+$ volatilization exerts a stronger buffering effect than carboxylic acid dissolution because $NH_{3T}$ concentrations are much larger.

## 3.2 Global distribution of rainwater pH and composition

Figure 6 (left panel) shows the global distribution of annual mean rainwater pH for the year 2013. Here we define 'rainwater' as all precipitation water (both rain and snow). The pH averaging procedure is as described in Sect. 2.3. The global mean





rainwater $\overline{pH}$ is 5.5 and varies from 4.5 over industrialized areas and the tropical forests to 8 over deserts, showing the same spatial patterns as cloudwater pH but with lower acidity because of dilution. Figure 7 shows the simulated concentrations of rainwater ions, except for $Na^+$ and $Cl^-$ which again do not contribute significantly to net acidity. Rainwater ion concentrations are on average 4 times more dilute than cloudwater concentrations (Fig. 3). The relative contribution of $SO_4^{2-}$ in industrialized regions is higher than for cloudwater because of additional $SO_4^{2-}$ from below-cloud scavenging of $SO_2$. The $NO_3^-$ contribution in central Africa is higher than for cloudwater because of high-altitude convective scavenging of $HNO_3$ produced from lightning $NO_x$. In Amazonia and tropical Africa, $HCOO^-$ and $CH_3COO^-$ contribute a larger fraction of rainwater acidity compared to cloudwater because of below-cloud scavenging, and as a result the rainwater pH is similar to that of cloudwater. Similarly, below-cloud scavenging of desert-generated dust results in alkaline rainwater over a much larger area compared to cloudwater.

The right panel in Fig. 6 shows the change in rainwater pH when the contribution from carboxylic acids is excluded. These acids biodegrade quickly, and thus their acidity is not generally captured by rainwater pH measurements. We find that rainwater pH increases by 0.4–1 unit in the Amazon, tropical Africa, and southeast Asia, consistent with observations (Andreae et al., 1990; Sanhueza et al., 1992; Sigha-Nkamdjou et al., 2003; Yoboué et al., 2005). Over the US, Europe, and eastern China the increase in pH is 0.1–0.4 units, which is similar to the observed contribution of carboxylic acids to rainwater $H^+$ (10–60%) in these areas (Keene and Galloway, 1984; Kawamura et al., 1996; Peña et al., 2002; Xu et al., 2010; Niu et al., 2018). Over the oceans, the change in pH from HCOOH and $CH_3COOH$ is small (~0.15 units) and in agreement with marine observations (Keene et al., 2015).

Figure 8 compares the simulated annual VWA rainwater pH ($\overline{pH_{VW}}$, Eq. 6) with observations from monitoring networks for the year 2013. Precipitation (rain and snow) pH observations are from the US National Trends Network (NTN) (NADP, 2019), the Canadian Air and Precipitation Monitoring Network (CAPMoN) (ECCC, 2018) , the European Monitoring and Evaluation Programme (EMEP) (EMEP, 2015), and the Acid Deposition Monitoring Network in East Asia (EANET) (EANET, 2019). We use monthly mean measurements from NTN (249 sites), EMEP (83 sites), and EANET (45 sites) and daily measurements from CAPMoN (30 sites), which we average to a monthly VWA $[H^+]$. The monthly $[H^+]$ values are used to calculate $\overline{pH_{VW}}$ following Eq. (6). The acidity from $HCOO^-$ and $CH_3COO^-$ is not accounted in the network measurements unless treated with biocide, which is done only in Malaysia. Thus, for comparison with observations, we use the simulated $\overline{pH_{VW}}$ calculated without $HCOO^-$ and $CH_3COO^-$.

GEOS-Chem rainwater pH values are largely consistent with the observations, reproducing the observed regional means to within 0.1 pH unit (observed: 5.26–5.30, GEOS-Chem: 5.23–5.39). Carboxylic acids lower the mean GEOS-Chem pH by 0.15–0.2 units. The model also generally reproduces the observed spatial variations within the regions. Observations and model show higher pH in the Midwest than in the rest of the US because of neutralization by agricultural $NH_3$. Rainwater pH in





southern Europe is also relatively high because of Saharan dust influence. Model values there are lower than in Fig. 6 because of different forms used to calculate annual mean pH (Eq. 6 here instead of Eq. 1). High [$SO_4^{2-}$] and [$NO_3^-$] and moderate neutralization lower the rainwater pH to below 4.5 in eastern China, Korea, and Japan. The high pH value observed in Xi'an (central China) is due to alkalinity from dust sources (EANET, 2016) but the corresponding dust influence in the model is

shifted slightly to the northwest. The low pH observed over Chongqing (central China) is because of high $SO_2$ and $NO_x$ emissions which are trapped locally by the surrounding terrain (Y. Chen et al., 2017).

Our global distribution of rainwater pH can be compared to previous model simulations by Rodhe et al. (2002) and Tost et al. (2007). Neither included carboxylic acids and thus they overestimated pH values over tropical continents. Tost et al. (2007)

did not include dust alkalinity either, resulting in large pH underestimates over desert regions. The pH values over eastern North America and Europe in these previous studies are about 0.5 units lower than in our simulation, reflecting the more recent decreases in $SO_2$ and $NO_x$ emissions (Hoesly et al., 2018).

### 3.3 Soil and freshwater acidification by wet deposition

Acidification of soil and freshwater is one of the major adverse effects of wet deposition fluxes on ecosystems because it

causes the leaching of nutrients, mobilizes toxic metals, and directly damages biota (Driscoll et al., 2001). Quantifying this effect requires accounting for post-depositional processes. The $H^+$ flux associated with carboxylates and $HCO_3^-$ is not relevant because carboxylic acids are readily consumed by bacteria, and the amount of $HCO_3^-$ in ecosystems is controlled by the ambient $CO_2$ concentrations (Reuss and Johnson, 1986). The acidifying effects of $NO_3^-$ and $NH_4^+$ depend on the biotic demand for nitrogen (N) (Reuss and Johnson, 1986). In ecosystems with high N demand (so-called N-limited ecosystems), $NO_3^-$ and $NH_4^+$

are readily assimilated by plants and microbes. Uptake of $NO_3^-$ is accompanied by the uptake of $H^+$ (or release of $OH^-$), cancelling the acidic effect of $NO_3^-$ deposition. $NH_4^+$ uptake is accompanied by the release of $H^+$, reversing the neutralizing effect of $NH_4^+$. Therefore, in N-limited ecosystems the acidic flux is calculated as follows (Rodhe et al., 2002):

$$F_{H^+(N-lim)} = F_{SO_4^{2-}} - F_{dust\ NVC} \tag{7}$$

where $F$ denotes the wet deposition flux in equivalents. However, in many industrialized regions, N deposition greatly exceeds

the biotic demand and results in N-saturated conditions (Aber et al., 1989; Watmough et al., 2005; Gundersen et al., 2006; Duan et al., 2016). In such conditions, only a small fraction of the deposited $NO_3^-$ and $NH_4^+$ is assimilated. The excess $NO_3^-$ causes $H^+$ accumulation, while the excess $NH_4^+$ can be converted by microbes to $NO_3^-$ (nitrification), which releases $2H^+$ for every $NH_4^+$ converted and also results in net $H^+$ formation. Considering the full acidifying potential of $NO_3^-$ and $NH_4^+$, we calculate the acidic flux in N-saturated conditions as follows (Galloway, 1995; Rodhe et al., 2002):

$$F_{H^+(N-sat)} = F_{SO_4^{2-}} + F_{NO_3^-} + F_{NH_4^+} - F_{dust\ NVC} \tag{8}$$

$F_{H^+(N-sat)}$ can be viewed as the upper limit of acidic inputs through wet deposition as some of the accumulated $NO_3^-$ can denitrify to $N_2$.



Figure 9 shows $F_{H^+(N\text{-lim})}$ and $F_{H^+(N\text{-sat})}$, along with the free $H^+$ flux which represents the direct acid input to ecosystems

excluding carboxylic acids. The global mean $F_{H^+(N\text{-lim})}$ over continents (4.1 meq m$^{-2}$ a$^{-1}$) is higher than the mean free $H^+$ flux

(3.1 meq m$^{-2}$ a$^{-1}$). The free $H^+$ flux is higher than $F_{H^+(N\text{-lim})}$ over central Africa and Amazonia because of $H^+$ associated with

$NO_3^-$ and $HCO_3^-$, respectively. $F_{H^+(N\text{-lim})}$ is highest over the eastern US, Central and South America, and East Asia, reflecting

high $SO_4^{2-}$ fluxes. The global mean $F_{H^+(N\text{-sat})}$ over continents (18 meq m$^{-2}$ a$^{-1}$) is much larger than $F_{H^+(N\text{-lim})}$ and the free $H^+$

flux because of acidity generated from $NH_4^+$ nitrification. Over eastern India, East and Southeast Asia, the eastern US, and

Central and South America, $F_{H^+(N\text{-sat})}$ is more than 50 meq m$^{-2}$ a$^{-1}$, which exceeds the critical load for acidification of highly

sensitive ecosystems with low acid buffering capacity (Kuylenstierna et al., 2001; Bouwman et al., 2002).

The total wet deposition of individual ions over the continents is largest for $NH_4^+$ ($1.3\times10^{12}$ eq a$^{-1}$) followed by $SO_4^{2-}$ ($1.0\times10^{12}$

eq a$^{-1}$) and $NO_3^-$ ($0.85\times10^{12}$ eq a$^{-1}$). Lamarque et al. (2013) reported that for the year 2000 the multi-model mean ($\pm$ standard

deviation) of the wet deposition flux over continents was higher for $SO_4^{2-}$ ($1.5 \pm 0.3 \times 10^{12}$ eq a$^{-1}$) than for $NH_4^+$ ($1.2 \pm 0.3 \times$

$10^{12}$ eq a$^{-1}$) and $NO_3^-$ ($1.1 \pm 0.2 \times 10^{12}$ eq a$^{-1}$). The decrease in the $SO_4^{2-}$ wet deposition flux between 2000 and 2013 reflects

the global decrease of anthropogenic $SO_2$ emissions (Hoesly et al., 2018). We find that $NH_4^+$ is now the largest source of

acidifying wet deposition, contributing 41% of $F_{H^+(N\text{-sat})}$ over continents globally, and it will contribute even more in the future

as global $NH_3$ emissions continue to grow (Hoesly et al., 2018).

**4 Conclusions**

We used the GEOS-Chem global model of atmospheric chemistry to simulate the global distributions of cloudwater and

rainwater acidity, and the total acid inputs from wet deposition to terrestrial ecosystems. This involved an improved pH

calculation in GEOS-Chem including contributions from dust alkalinity, sea salt aerosol alkalinity, and carboxylic acids

(HCOOH and $CH_3COOH$). Our prime motivation was to better understand and evaluate the global cloudwater pH distribution

in the model for future simulations of sulfate, organic, and halogen chemistry. Extending the analysis to rainwater pH provided

further opportunity for model evaluation and allowed us to quantify post-depositional effects in acid inputs to ecosystems on

a global scale.

We compiled cloudwater pH measurements worldwide from the literature and compared them to the GEOS-Chem simulation.

The global mean cloudwater pH is $5.2 \pm 0.9$ in the observations, and $5.0 \pm 0.8$ in GEOS-Chem sampled at the same locations.

The lowest pH values of 3–4 are over East Asia because of high acid inputs and despite an average 70% neutralization by $NH_3$

and dust cations. Low pH values extend across the North Pacific because of weak neutralization. Cloudwater pH is 4–5 over

the US and Europe with dominant acid input from $HNO_3$ and over 50% neutralization from $NH_3$. Alkaline cloudwater with



pH as high as 8 is found over the northern subtropical desert belt extending from the Atlantic Ocean to Mongolia, including

western India. Carboxylic acids account for less than 25% of the cloudwater $H^+$ in the northern hemisphere, but 25–50% in the southern hemisphere and over 50% in the southern tropical continents where they drive the pH to below 4.5. We find little dependence of cloudwater pH on altitude other than dilution from changes in liquid water content.

Anthropogenic emissions of $SO_2$ and $NO_x$ are decreasing rapidly in the developed world, and this together with the large

fraction of neutralized acidity might be expected to lead to large increases in cloudwater pH. However, there is a strong buffering effect because of the semi-volatility of $NH_4^+$ and carboxylates. We find that a factor of 2 decrease in $SO_4^{2-}$ and $NO_3^-$ inputs over the US increases the cloudwater pH by 1 unit, compared to an increase of 2.1 units in the absence of buffering.

The global mean rainwater pH in GEOS-Chem is 5.5, higher than cloudwater pH because of dilution and below-cloud

scavenging of bases ($NH_3$, dust). Rainwater pH shows spatial patterns similar to cloudwater pH but is influenced more strongly by carboxylic acids and dust near source regions because of below-cloud scavenging. GEOS-Chem is consistent with the annual mean rainwater pH observed at monitoring networks in North America, Europe, and East Asia. We find that the carboxylic acids lower the rainwater pH by up to 1 unit in the Amazon, tropical Africa, and southeast Asia, and by about 0.2 units in the US, Europe, and East Asia. This pH depression would not be seen in the observations because of fast biological

consumption of the carboxylic acids after deposition.

Lastly, we examined the total acid inputs to soil and freshwater from wet deposition, including the post-depositional effects from $NH_4^+$ and $NO_3^-$ utilization by the biosphere. We find that total acid inputs under N-saturated conditions exceed 50 meq $m^{-2}$ $a^{-1}$ in many parts of East Asia and the Americas, a level that can damage sensitive ecosystems. $NH_4^+$ contributes 41% of

the acid input under N-saturated conditions globally.



*Data availability:* The NTN data is available at http://nadp.slh.wisc.edu/data/NTN/ntnAllsites.aspx (last access: October 11, 2019), the CAPMoN data at http://donnees.ec.gc.ca/data/air/monitor/monitoring-of-atmospheric-precipitation-chemistry/?lang=en (last access: October 18, 2019), the EMEP data at https://projects.nilu.no//ccc/emepdata.html (last access: April 1, 2020), and the EANET data at https://monitoring.eanet.asia/document/public/index (last access: October 9, 2019). The model results are available on request from the corresponding author.

*Author contributions:* VS and DJJ designed the study. VS carried out the modeling and analysis. JMM, XW, and SZ contributed model updates. VS and DJJ wrote the paper with input from all authors.

*Competing interests:* The authors declare that they have no conflict of interest.

*Acknowledgements:* We thank the US National Trends Network, the Canadian Air and Precipitation Monitoring Network, the European Monitoring and Evaluation Programme, and the Acid Deposition Monitoring Network in East Asia for the rainwater pH measurements.

*Financial support:* This work was supported by the NASA Atmospheric Chemistry Modeling and Analysis Program, by the Atmospheric Chemistry Program of the US National Science Foundation, and by the US EPA Science To Achieve Results (STAR) program.





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



**Table 1: Henry's law coefficients for the calculations of cloud- and rainwater pH[a]**

| Species | $H$ (M atm$^{-1}$) at 298 K | $\frac{d \ln H}{d(1/T)}$ (K) |
|---|---|---|
| HNO$_3$ | $2.1 \times 10^5$ | 0 |
| HCl | $1.5 \times 10^3$ | 2300 |
| NH$_3$ | 60 | 4200 |
| HCOOH | $8.8 \times 10^3$ | 6100 |
| CH$_3$COOH | $4.0 \times 10^3$ | 6300 |
| SO$_2$ | 1.2 | 3100 |
| CO$_2$ | $3.4 \times 10^{-2}$ | 2400 |

[a] From the compilation of Sander (2015).





**Table 2: Acid/base dissociation constants for the calculations of cloud- and rainwater pH[a]**

| Equilibrium reactions | $K$ (M or M$^2$) at 298 K | $\frac{d \ln K}{d(1/T)}$ (K) |
|---|---|---|
| $HNO_3$ (aq) $\leftrightarrow$ $H^+ + NO_3^-$ | 15 | 8700 |
| $HCl$ (aq) $\leftrightarrow$ $H^+ + Cl^-$ | $1.7 \times 10^6$ | 6900 |
| $NH_4OH$ (aq) $\leftrightarrow$ $NH_4^+ + OH^-$ | $1.7 \times 10^{-5}$ | -450 |
| $HCOOH$ (aq) $\leftrightarrow$ $H^+ + HCOO^-$ | $1.8 \times 10^{-4}$ | 150 |
| $CH_3COOH$ (aq) $\leftrightarrow$ $H^+ + CH_3COO^-$ | $1.7 \times 10^{-5}$ | 50 |
| $SO_2 \cdot H_2O$ $\leftrightarrow$ $H^+ + HSO_3^-$ | $1.3 \times 10^{-2}$ | 2000 |
| $HSO_3^-$ $\leftrightarrow$ $H^+ + SO_3^{2-}$ | $6.6 \times 10^{-8}$ | 1500 |
| $CO_2 \cdot H_2O$ $\leftrightarrow$ $H^+ + HCO_3^-$ | $4.3 \times 10^{-7}$ | -1000 |
| $HCO_3^-$ $\leftrightarrow$ $H^+ + CO_3^{2-}$ | $4.7 \times 10^{-11}$ | -1800 |
| $H_2O$ $\leftrightarrow$ $H^+ + OH^-$ | $1 \times 10^{-14}$ | -6700 |
| $CaCO_3$(s) $\leftrightarrow$ $Ca^{2+} + CO_3^{2-}$ | $3.3 \times 10^{-9}$ | -1200 |

[a] From Pandis and Seinfeld (1989), except for $CaCO_3$(s) (Nordstrom et al., 1990), and HCOOH(aq) and CH$_3$COOH(aq) (Khare et al., 1999).



**Table 3: Species included in the cloudwater pH calculation[a]**

| Conserved totals ≡ Sum of partitioned species |
| --- |
| $H_2SO_{4,T} \equiv SO_4^{2-}$ [b] |
| $HNO_{3,T} \equiv HNO_3(g) + HNO_3(aq) + NO_3^-$ |
| $HCl_T \equiv HCl(g) + HCl(aq) + Cl^-$ |
| $NH_{3,T} \equiv NH_3(g) + NH_4OH(aq) + NH_4^+$ |
| $HCOOH_T \equiv HCOOH(g) + HCOOH(aq) + HCOO^-$ |
| $CH_3COOH_T \equiv CH_3COOH(g) + CH_3COOH(aq) + CH_3COO^-$ |
| $SO_{2,T} \equiv SO_2(g) + SO_2(aq) + HSO_3^- + SO_3^{2-}$ |
| $CO_{2,T} \equiv CO_2(g) + CO_2(aq) + HCO_3^- + CO_3^{2-}$ [c] |
| $Ca_T \equiv Ca^{2+} + CaCO_3(s)$ |
| $Na_T \equiv Na^+$ |

[a] The calculation assumes a closed system for the cloudy fraction of the model grid cell where concentration totals ($T$) are conserved and are partitioned between species using the Henry's law and acid-base dissociation equilibria of Tables 1 and 2, and the local cloudwater liquid water content and temperature.

[b] $H_2SO_4$ has sufficiently low vapor pressure to be completely in the cloudwater phase, and $H_2SO_4(aq)$ and $HSO_4^-$ concentrations are negligible at typical cloudwater pH ($> 3$).

[c] $CO_2(g)$ mixing ratio is taken to be 390 ppm as representative of 2013.





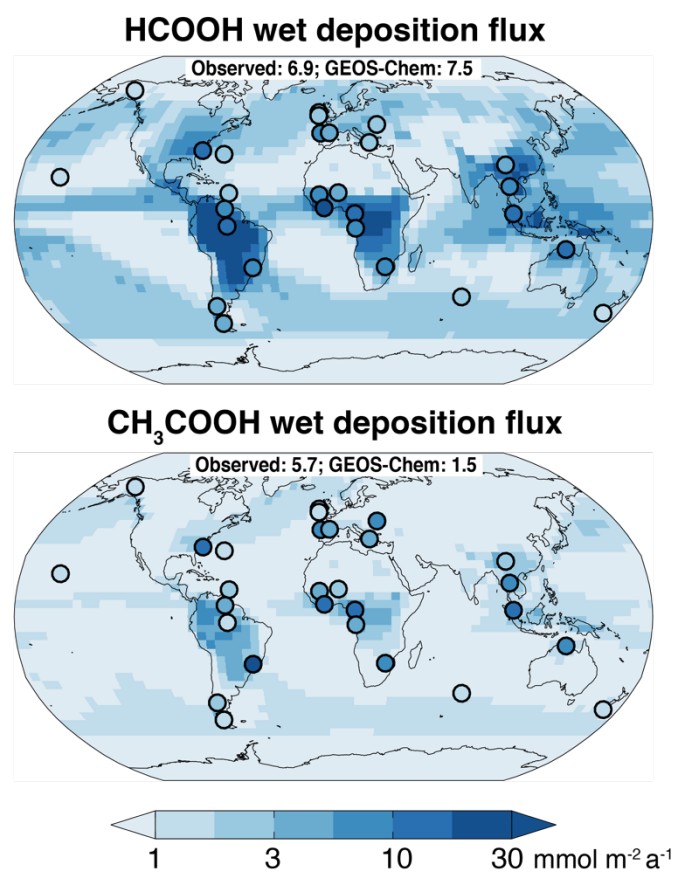

**Figure 1: Annual mean wet deposition fluxes of HCOOH and CH₃COOH.** The GEOS-Chem model values (contours) are for the year 2013 and the observations (circles) are for various years as compiled by Vet et al. (2014) and Keene et al. (2015). The compilations only include studies that used adequate methods to preserve HCOOH and CH₃COOH in rainwater samples. We exclude studies with measurement periods of less than a year. For studies that reported only rainwater concentrations, we estimate the deposition fluxes using climatological rainfall data for the corresponding locations from the Global Precipitation Climatology Center (Meyer-Christoffer et al., 2018). The global mean observed fluxes and the corresponding GEOS-Chem fluxes are shown inset.

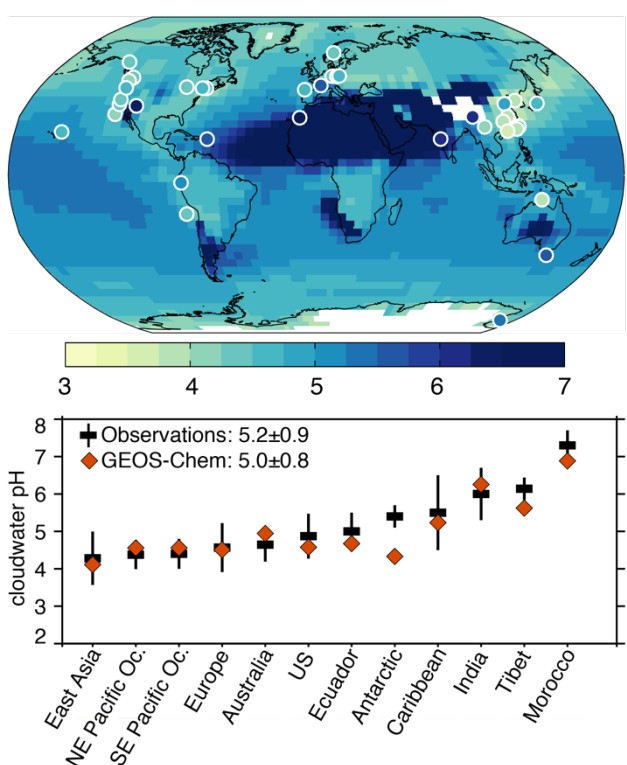

**Figure 2: Observed and simulated cloudwater pH. The top panel shows the GEOS-Chem annual mean cloudwater pH below 700 hPa for the year 2013 along with cloudwater pH observations (filled circles) collected since 1980 (Table A1). See Sect. 2.3 for the procedure to compute average pH in the model. White color denotes areas where the topographic elevation is higher than 700 hPa. The maximum modeled and observed pH values are 8.2 and 7.3, respectively. The bottom panel shows the observations grouped by regions (Table A1), with means ± standard deviations calculated from the ensemble of data sets for the region, and the corresponding GEOS-Chem mean values sampled at the location and month of the measurements. The global mean ± standard deviation pH values computed from the regional mean observed and modeled values are inset in the bottom panel.**

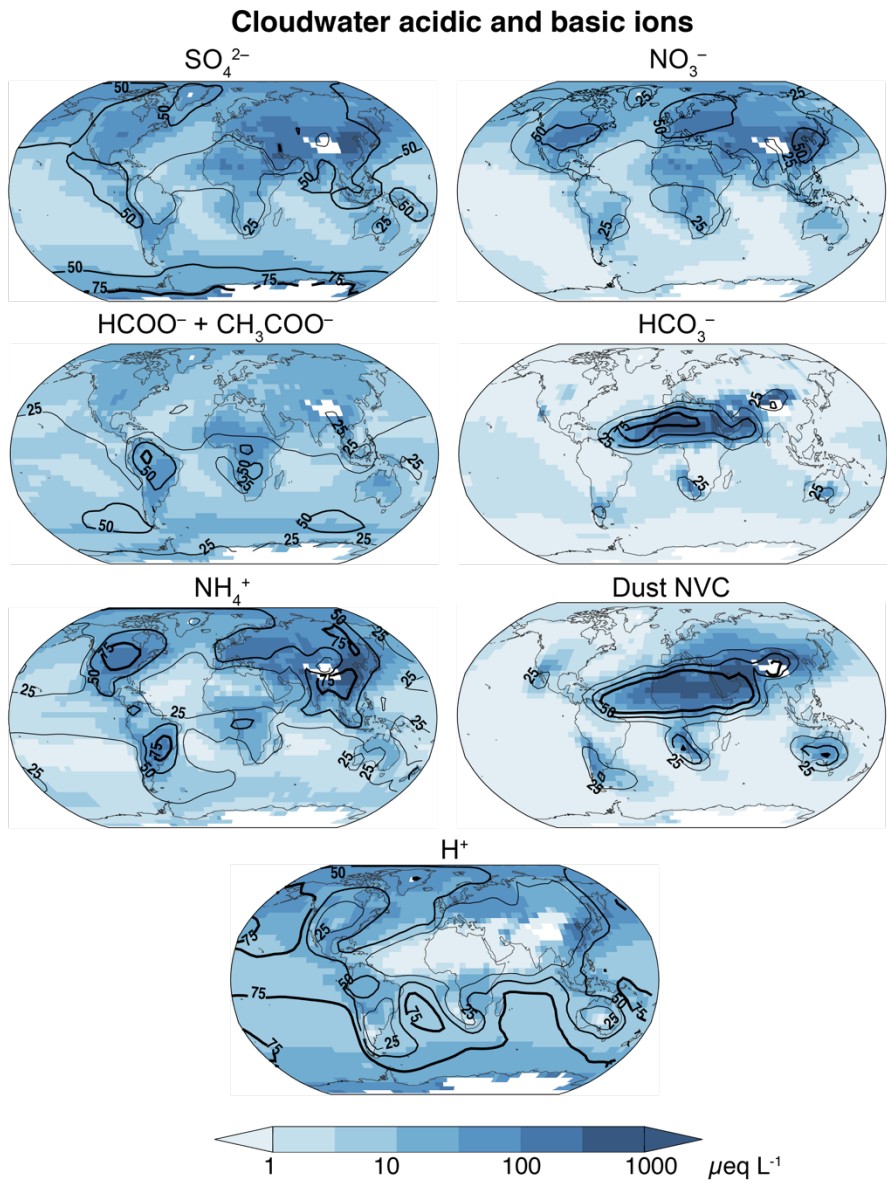

**Figure 3:** GEOS-Chem cloudwater equivalent concentrations of major acidic and basic ions. Values are annual volume weighted averages below 700 hPa for the year 2013. The contour lines (25%, 50%, 75%) show the percent contribution of each ion to the total anion or cation equivalents. Na⁺ and Cl⁻ are not included in this total because they are largely in balance and make little net contribution to acidity. White color denotes areas where the topographic elevation is higher than 700 hPa.



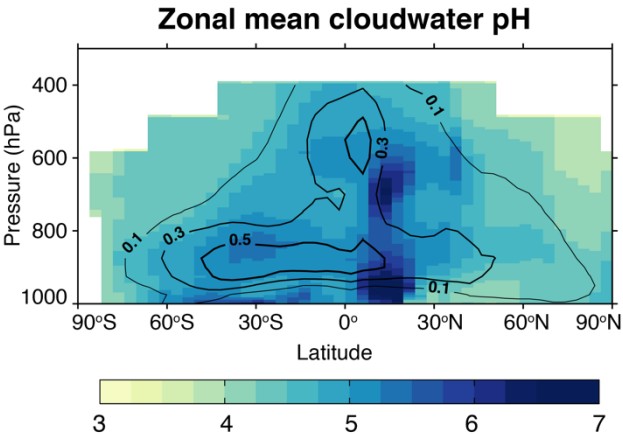

**Figure 4: Zonal mean cloudwater pH simulated by GEOS-Chem. Values are annual volume weighted averages for the year 2013. The contour lines show the annual mean MERRA-2 cloud liquid water content in g m⁻³ for the cloudy fraction of grid cells where a liquid cloud is present. White color denotes areas where the cloud liquid water content is below 0.01 g m⁻³.**





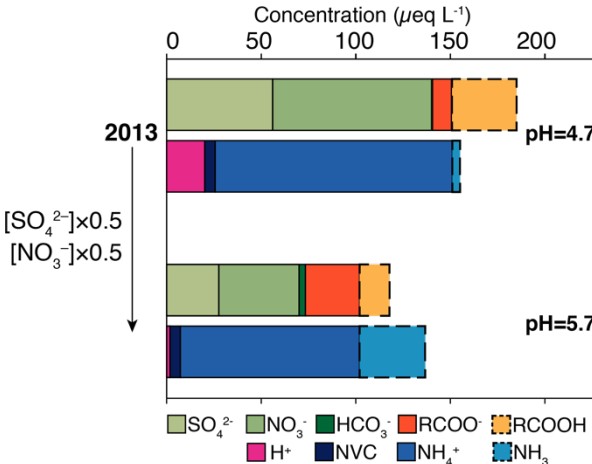

**Figure 5: Sensitivity of cloudwater pH to decreasing acid inputs over the contiguous US. The upper bars show the volume weighted average concentrations of acidic and basic ions for 2013, with additional dashed lines showing the undissociated concentrations of carboxylic acids (RCOOH) and the gas-phase concentration of ammonia (NH₃). The lower bars show the effect of decreasing [SO₄²⁻] and [NO₃⁻] by half relative to 2013 levels. All concentrations are expressed as cloudwater equivalents. The corresponding pH values are indicated.**



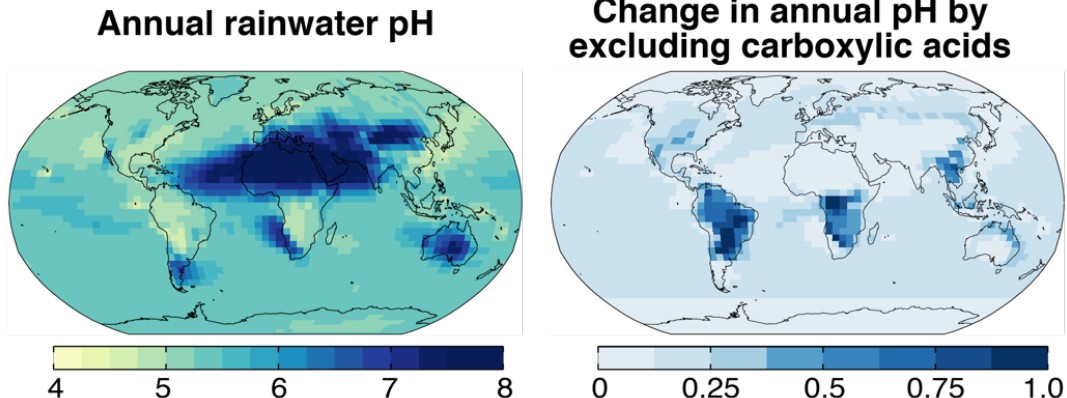

**Figure 6: Annual mean rainwater pH simulated by GEOS-Chem for 2013. The right panel shows the change in rainwater pH when [HCOO⁻] and [CH₃COO⁻] are excluded from the ionic charge balance. Rainwater as defined here includes snow as well as rain. See Sect. 2.3 for the procedure to compute average pH in the model.**



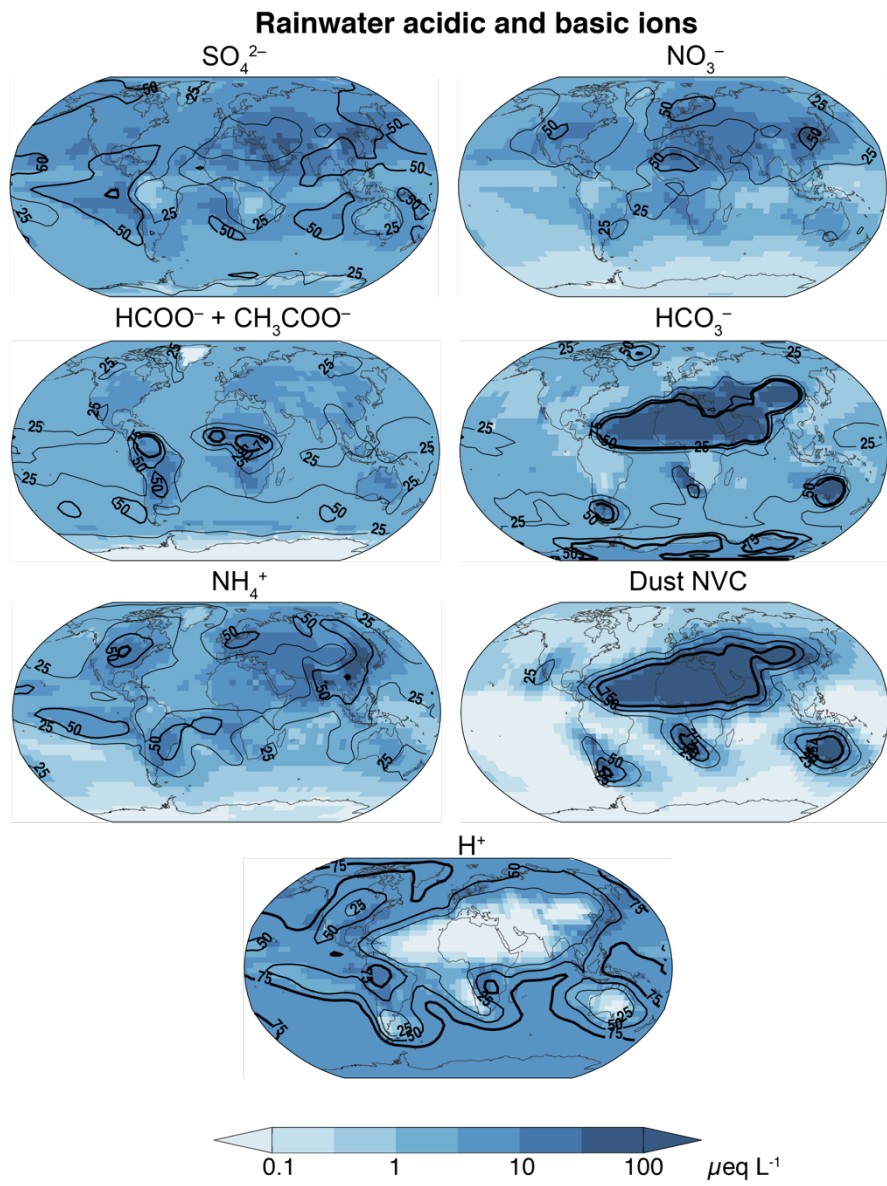

**Figure 7: GEOS-Chem rainwater equivalent concentrations of major acidic and basic ions for the year 2013. The contour lines (25%, 50%, 75%) show the percent contribution of each ion to the total anion or cation equivalents. Na$^+$ and Cl$^-$ are not included in this total because they are largely in balance and make little net contribution to acidity.**





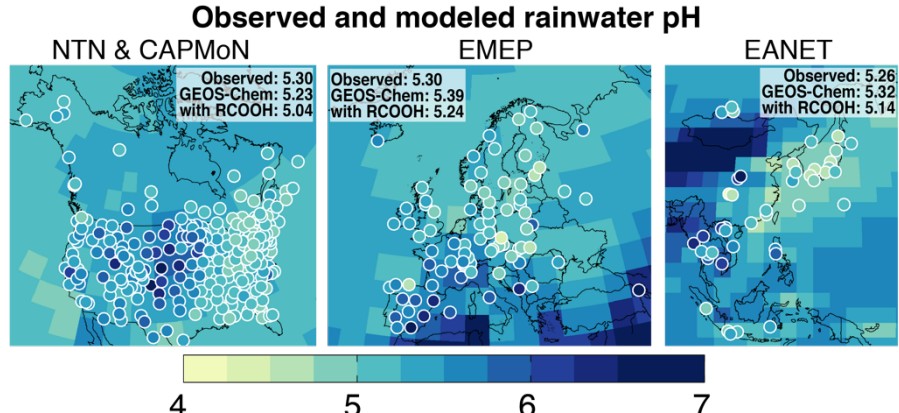

**Figure 8: Annual volume weighted average (VWA) rainwater pH over North America, Europe and East Asia for the year 2013. Values are the precipitation volume weighted averages of the monthly means (Eq. 6). The GEOS-Chem model values (solid background) exclude RCOOH (HCOOH and CH$_3$COOH) in the pH calculation for comparison to observations. Observations (circles) are from the US National Trends Network (NTN), the Canadian Air and Precipitation Monitoring Network (CAPMoN), the European Monitoring and Evaluation Programme (EMEP), and the Acid Deposition Monitoring Network in East Asia (EANET). The EANET sites in Malaysia are not shown because of their different sampling procedure (see text). The insets in each panel show the spatial means of the observations for the corresponding region together with the corresponding GEOS-Chem means at the measurement locations. The mean GEOS-Chem pH values with RCOOH included in the pH calculations are also shown.**

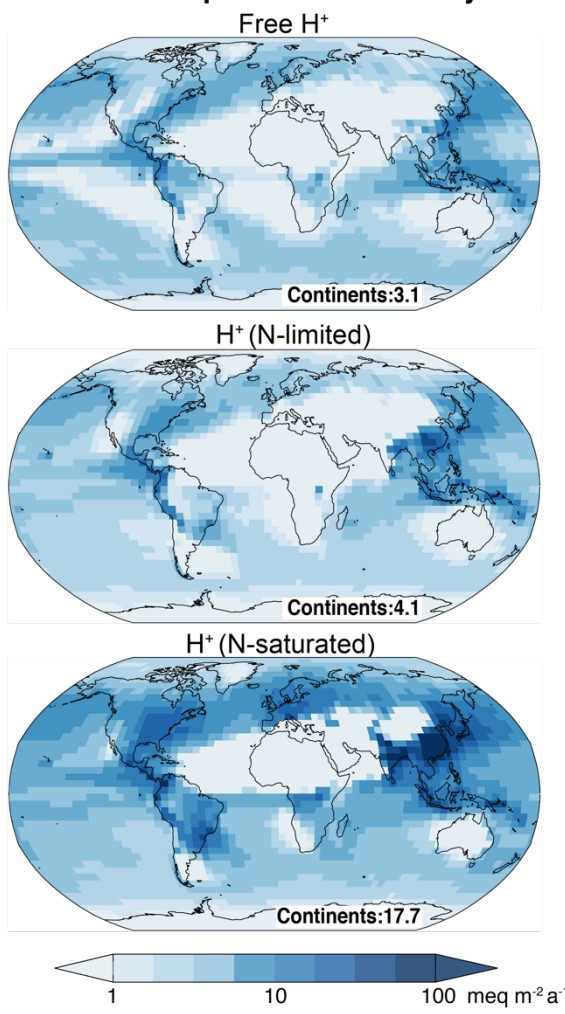

**Figure 9: Acid wet deposition fluxes including post-depositional effects.** Values are annual means for 2013 as simulated by GEOS-Chem. H⁺ associated with carboxylic acids is excluded as these acids are rapidly consumed by bacteria. The free H⁺ flux represents the direct H⁺ input to ecosystems not accounting for any post-deposition nitrogen (N) transformation. The H⁺ flux for N-limited conditions is computed using Eq. (7) and assumes complete assimilation of $NO_3^-$ and $NH_4^+$, while the H⁺ flux for N-saturated conditions is computed using Eq. (8) and assumes complete nitrification of $NH_4^+$ and no assimilation of $NO_3^-$. The global mean flux over continents in units of meq m⁻² a⁻¹ is indicated.



## Appendix

**Table A1: Cloudwater pH observations grouped by region and listed from west to east and north to south.**

| Location | Altitude (km) | Period | Mean ± std dev | Median | Range | Reference |
|---|---|---|---|---|---|---|
| **NE Pacific Ocean** | | | | | | |
| California coast 29-33N; 121-123W | 0–1 | Jul, 2001 | 4±0.4 [a, b] | 4 | 3.3–4.8 | Straub et al. (2007) |
| California coast 35-37N; 122-123W [a] | 0.12–0.8 | Jul–Aug, 2011 | 4.5±0.7 | - | - | Z. Wang et al. (2014) |
| California coast 34-43N; 119-126W | 0.12–1 | Jul–Aug, 2013 | 4.3±0.5 | - | - | Prabhakar et al. (2014) |
| Hawaii 22N; 152W | 0–1 | Jun, 1980 | 4.5±0.1 [a, c] | - | 4.2–4.7 | Parungo et al. (1982) |
| Alaska, US 58N; 135W | 0.8 | Aug–Sep, 1984; July–Aug, 1985 | 4.5±0.4 | 4.8 | - | Bormann et al. (1989) |
| Whistler Mtn, Canada, 50N;123N | 1.7 | Jun–Jul, 2010 | 4.4 [b] | 4.4 | - | Ervens et al. (2013) |
| Cheeka Peak, US, 48N; 125W | 1 | May, 1993 | 4.2±0.2 | - | - | Vong et al. (1997) |
| Mary's Peak, US, 45N; 124W | 1.3 | Jul–Nov, 1985 | 4.7±0.3 [a] | 5.2 | 4.4-4.9 | Bormann et al. (1989) |
| **Continental United States** | | | | | | |
| Mt. Washington 44N; 71W | 1.9 | Jun–Aug, 2008–2010 | 4.3 | - | - | G. Murray et al. (2013) |
| Whiteface Mountain 44N; 74W | 1.5 | Jun–Sep, 2010–2013 | 4.5±0.8 [a] | - | 3.4-6.6 | Schwab et al. (2016) |
| Michigan 44-46N; 83-85W | 1.4–3.1 | Jul–Aug, 2005 | 4.4±0.8 [a] | - | 2.2-5.2 | Hill et al. (2007); Pye et al. (2020) |
| Mt. Elders 35N; 112W | 2.8 | Jun–Sep, 2005–2007 | 6.3±0.4 [a] | - | 5.1-6.6 | Hutchings et al. (2009) |
| **Caribbean** | | | | | | |
| Puerto Rico 18N; 65W | 1.1 | 2004–2007 | 5.5±1.0 | - | - | Gioda et al.(2011) |
| **Southeast Pacific Ocean** | | | | | | |
| Peru/Chile coast 16-24S; 72-82W | 0–1 | Oct–Nov,2008 | 4.4±0.4 [a] | - | 3.5-6 | Benedict et al. (2012) |





| Location | Altitude (km) | Period | Mean ± std dev | Median | Range | Reference |
|---|---|---|---|---|---|---|
| **Ecuador** | | | | | | |
| Andes 79W; 4S | 1.9–3.2 | 2004–2009 | 5.0±0.5 [a] | - | - | Makowski Giannoni et al. (2013, 2016); Pye et al. (2020) |
| **Europe** | | | | | | |
| Åreskutan, Sweden, 63N; 13E | 1.3 | Jul–Aug, 1983; Jul–Aug, 1984 | 4.4 | - | - | Ogren & Rodhe (1986) |
| Mt. Schmucke, Germany, 51N; 11E | 0.9 | Sep–Oct, 2010 | 4.3±0.4 [a] | 4.6 | 3.6-5.3 | van Pinxteren et al. (2016) |
| Mt. Milesovka, Czech Rep., 51N; 14E | 0.8 | May–June 2006 | 4.1±0.2 [a] | - | 3.8-4.7 | Fisak et al. (2009) |
| Mt. Szrenica, Poland, 51N; 16E | 1.3 | Dec, 2005 – Dec, 2006 | 4.6±1.0 [a] | - | 3.5-7.4 | Błaś et al. (2010) |
| Niesen, Switzerland, 47N; 7-8E | 1.6–2.3 | Apr–Oct, 2006; Apr–Oct 2007 | 6.5±0.5 [a] | - | 5.8-7.7 | Michna et al. (2015) |
| Puy de Dome, France, 46N; 3E | 1.5 | 2001–2006; 2009–2011 | 5.5±1.1 [a] | 5.6 | 3.1-7.6 | Deguillaume et al. (2014) |
| Xistral Mountains, Spain, 44N; 8W | 0.9 | Sep, 2011–Apr 2012 | 4.5±0.4 [a] | - | 3.8-5.2 | Fernández-González et al. (2014) |
| **Morocco** | | | | | | |
| Mt. Boutmezguida 29N; 10W | 1.2 | Nov–Jun, 2013–2015 | 7.3±0.4 [a] | - | 7–8.5 | Schunk et al. (2018); Pye et al. (2020) |
| **India** | | | | | | |
| Sinhagad 18N; 74E | 1.5 | 2007–2010 | 6.0±0.7 [a] | - | 4.7–7.4 | Budhavant et al. (2014) |
| **Tibet** | | | | | | |
| Sejila mountain 30N; 95E | 4 | Jul, 2017–Sep, 2018 | 6.1±0.3 | - | - | W. Wang et al. (2019) |
| **East Asia** | | | | | | |
| Yellow Sea 38N;125E | 0.1 | Jul, 2014 | 3.9±0.4 [a] | - | 3.5–5 | Boris et al. (2016); Pye et al. (2020) |
| Mt. Tateyama, Japan, 37N,138E | 2.5 | Sept–Oct, 2007–2009 | 4.5±0.7 [a] | - | 3.5–6.3 | K. Watanabe et al. (2010) |





| Location | Altitude (km) | Period | Mean ± std dev | Median | Range | Reference |
|---|---|---|---|---|---|---|
| **East Asia (contd.)** | | | | | | |
| Mt. Tai, China, 36N; 117E | 1.6 | Jul–Oct, 2014 | 5.9±0.8 [a] | - | 3.8-7.0 | J. Li et al. (2017) |
| | | Mar–Apr, June–July, Oct–Nov, 2007; Mar–Apr, Jun–Jul, 2008 | 4.3±1.3 [a] | - | 2.6-7.6 | Guo et al. (2012) |
| | | Jun–Aug 2015 | 4.9±0.6 [a] | - | 3.8–6.3 | Zhu et al. (2018) |
| Mt. Lu, China, 30N; 116E | 1.2 | Aug–Sep, 2011; Mar–May, 2012 | 3.8±0.7 [a] | - | 2.8–5.6 | L. Sun et al. (2015) |
| Mt. Heng, China, 27N; 113E | 1.3 | Mar–May, 2009 | 3.8±1.0 [a] | - | 2.9–6.9 | M. Sun et al. (2010) |
| Ailaoshan, China, 25N; 101E | 2.5 | Dec 2015–Mar 2016 | 4.1±0.4 [a] | - | 3.5–4.9 | Nieberding et al. (2018) |
| Mt. Bamboo, Taiwan, 25N,122E | 1.1 | Jan–Mar 2009 | 4.1±0.6 [a] | - | 3.1–5.6 | Sheu & Lin (2013) |
| Chilan Mtn, Taiwan, 25N,122E | 1.7 | Apr–May, 2011 | 4.5±0.4 [a] | - | 3.7–5.2 | Simon et al. (2016) |
| Lulin Station, Taiwan, 23N,121E | 2.9 | Apr–May, 2011 | 3.9±0.3 [a] | - | 3.4–4.5 | Simon et al. (2016) |
| Mt. TaiMoSha, Hong Kong, 22N;114E | 1 | Oct–Nov 2016 | 3.6±0.7 [a] | - | 3.0–6.0 | Li et al. (2020) |
| **Australia** | | | | | | |
| Cape Grim, 40-42S; 144-149E | 0.6–1.5 | Jun, 1981; Mar, 1983 | 5.5±0.5 | - | - | Gillett & Ayers (1989) |
| Jabiru, 133E; 13S | 2.7–3.7 | Nov, 1985 | 3.8±0.4 [a] | - | 3.5–5.2 | Ayers & Gillett (1988) |
| **Antarctic** | | | | | | |
| Antarctica coast 78S; 167E | 0.6–1.5 | Dec, 1982 | 5.4±0.3 [a] | - | 4.9–6.2 | Saxena & Lin (1990) |

[a] Standard deviation estimated as range / 4.

[b] Median value used as the mean.

[c] Mean estimated as (max+min) / 2.