# Peer review of "Global modeling of cloudwater acidity, precipitation acidity, and acid inputs to ecosystems"

_Atmospheric Chemistry and Physics, 2020_

## Referee Comment (RC1) · Anonymous Referee #1 · 16 Jul 2020

Shah et al. report global predictions of cloud and precipitation acidity from an updated version of GEOS-CHEM. The addition of formic and acetic acids and of dust enable the model to better capture regionally important influences on hydrometeor pH and to better match spatial variations in acidity relative to other recent model predictions. The work is thorough, carefully conducted and, for the most part, clearly explained. I have a few comments and suggestions:

1. abstract, lines 21-22: please clarify that the model successfully reproduces annual mean rainwater pH observations.

2. The authors refer to rainwater acidity and go to the trouble of somewhat strangely defining rainwater to include snow. Why not just use the more general term "precipitation" throughout?

[Figure]

3. The addition of formic and acetic acids captures important influences on cloud and precipitation acidity. a. it would be useful to mention that future work might also warrant consideration of other key dicarboxylic acids (e.g., oxalic, succinic) that can be abundant in clouds. b. I was disappointed to see the authors recognize their new scheme continues to significantly underpredict acetic acid concentrations, as evidenced by comparison to source strengths considered by others as well as observed wet deposition fluxes (a factor of 4) but did not take action to correct for this discrepancy. They point out that a biogenic emissions scaling could correct this discrepancy similar to how they corrected for HCOOH underpredictions. Why not make that correction as part of this paper?

4. Eq'n. 1: The authors exclude both $CO_3^{2-}$ and $OH^-$ from this charge balance equation, arguing that both are small for pH<8. I am OK with this omission of $CO_3^{2-}$ but worry about artifacts in their higher pH simulations due to omission of $OH^-$. At pH 7, for example $[OH^-]$ equals $[H^+]$, meaning that omission of $OH^-$ introduces a relatively significant error in the $H^+$ concentration obtained from eq'n 1 in this pH range. Why not just include $OH^-$ and eliminate this high pH artifact? How does this artifact of $OH^-$ omission at high pH influence the comparison with other cloud models at high pH (e.g., p.9, lines 59+)

5. The authors discuss the importance of $NH_4^+$ and carboxylic acid buffering on cloud pH when strong acid inputs decrease. It would be good to mention here that other buffers may also play an important role, including $HCO_3^-$ (see, e.g., Collett et al. (1999) Internal acid buffering in San Joaquin Valley fog drops and its influence on aerosol processing. Atmos. Environ., 33, 4833-4847).

6. p. 13, lines 89-92: Again talking about the pH buffering effect, the authors point out that a factor of 2 decrease in sulfate and nitrate inputs yields a pH increase of 1 unit, with buffering, vs. 2.1 units predicted w/o buffering. Have they examined historical time series of cloud pH to see if this effect is apparent in ambient observations? One simple comparison to ballpark this might be to look at the pH trends over time in various

regions of the world as summarized by Pye et al. (2020) vs. known changes in NOx and SOx emissions in those regions.

7. The manuscript would benefit from a more detailed discussion of how cloud pH data were selected for comparison to model predictions. The authors include several observational studies in a number of regions, but certainly not all. I understand that some points are omitted due to source proximity and some, I think, were omitted due to being from an era far from the 2013 simulation and its emissions levels (although some fairly old measurements are included). Others (e.g., several studies from Japan mountain sites and the recent Kim et al. observations from the northern Pacific (https://doi.org/10.1007/s10874-020-09403-8)) are not included. A more robust discussion of why various datasets were included/excluded would give the reader greater confidence in the very promising matches shown with modeled pH values by region.
* * *

---

## Referee Comment (RC2) · Anonymous Referee #2 · 22 Jul 2020

Review of "Global modeling of cloudwater acidity, rainwater acidity, and acid inputs to ecosystems" by Shah et al.

This is a modeling study aiming to present a global view of cloud and rainwater pH, with particularly important aspects of the work including improved cloudwater pH calculations, inclusion of some carboxylic acids and dust alkalinity, and an explicit rainwater pH calculation. The topic is of relevance to the journal and addresses and important issue in the field of atmospheric sciences. I found the paper to be very well written, clear, and with conclusions well supported by the analysis. I recommend publication and do not have any additional suggestions for the authors.

[Figure]

2020.

---

## Author Comment (AC1) · 27 Aug 2020

**We thank the referees for their insightful comments. Our point-by-point responses to the comments and the revisions to the manuscript are as follows:**

**Anonymous Referee #1**

Received and published: 16 July 2020

Shah et al. report global predictions of cloud and precipitation acidity from an updated version of GEOS-CHEM. The addition of formic and acetic acids and of dust enable the model to better capture regionally important influences on hydrometeor pH and to better match spatial variations in acidity relative to other recent model predictions. The work is thorough, carefully conducted and, for the most part, clearly explained. I have a few comments and suggestions:

1. abstract, lines 21-22: please clarify that the model successfully reproduces annual mean rainwater pH observations.

We have clarified in the abstract of the revised manuscript that the model reproduces annual mean observations.

2. The authors refer to rainwater acidity and go to the trouble of somewhat strangely defining rainwater to include snow. Why not just use the more general term "precipitation" throughout? Following the referee's suggestion, we now use the term "precipitation" instead of "rainwater" throughout the manuscript.

3. The addition of formic and acetic acids captures important influences on cloud and precipitation acidity. a. it would be useful to mention that future work might also warrant consideration of other key dicarboxylic acids (e.g., oxalic, succinic) that can be abundant in clouds. b. I was disappointed to see the authors recognize their new scheme continues to significantly underpredict acetic acid concentrations, as evidenced by comparison to source strengths considered by others as well as observed wet deposition fluxes (a factor of 4) but did not take action to correct for this discrepancy. They point out that a biogenic emissions scaling could correct this discrepancy similar to how they corrected for HCOOH underpredictions. Why not make that correction as part of this paper?

We did not attempt to correct the model underestimate of CH3COOH as its effect on pH is small. We have clarified this further and emphasized the need to better understand the sources of carboxylic acids with the following additions:

On page 5, line 18 (underlined part added):

"model CH3COOH could be corrected similarly to HCOOH in future work by scaling up biogenic emission. But the effect is relatively small. We find in the model that the global mean cloudwater pH would decrease by 0.05 units if we increased CH3COOH concentrations by a factor of 4."

And on page 13, line 24 we have added:

"Carboxylic acids affect cloudwater and precipitation pH globally, but their sources are uncertain. Our simulation could reproduce the observed HCOOH wet deposition flux by using scaled-up biogenic emissions but underestimated CH3COOH flux by a factor of 4, indicating that a better understanding of their sources is needed. Dicarboxylic acids, such as oxalic, succinic, and malonic acids, are also present in cloudwater and precipitation and their effect on pH needs to be evaluated." 4. Eq'n. 1: The authors exclude both CO32- and OH- from this charge balance equation, arguing that both are small for pH<8. I am OK with this omission of CO32- but worry about artifacts in their higher pH simulations due to omission of OH-. At pH 7, for example [OH-] equals [H+], meaning that omission of OH- introduces a relatively significant error in the H+ concentration obtained from eq'n 1 in this pH range. Why not just include OH- and eliminate this high pH artifact? How does this artifact of OH- omission at high pH influence the comparison with other cloud models at high pH (e.g., p.9, lines 59+)

The omission of [OH-] in Eq. (1) incurs a negligible bias. We have clarified this in the text on page 9 as follows (underlined text added):

" $[HCO_3^-]$  is calculated from equilibrium with atmospheric CO2 as follows:

$$\overline{[\text{HCO}_3^-]} = \frac{H_{\text{CO}_2} K_{\text{c1}} P_{\text{CO}_2}}{\overline{[\text{H}^+]}}$$
(4)

where  $H_{CO_2}$  and  $K_{c1}$  are the Henry's law coefficient for CO2 and the CO2(aq)/HCO3- acid dissociation constant, respectively, at the average cloudwater temperature for the period and domain (Tables 1 and 2).  $P_{CO_2}$  is the CO2 partial pressure, taken to be 390 ppm as representative of 2013. Substituting these values in Eq. (4), [HCO3-]  $\approx 10^{-11.3}/[H^+]$ . Over the range of cloudwater pH values (3–8.5), [CO32-] ( $\approx 10^{-21.6}/[H^+]^2$ ) and [OH-] ( $\approx 10^{-14}/[H^+]$ ) are negligible compared to [HCO3-] and omitted from Eq. (1) (Stumm et al., 1987)."

5. The authors discuss the importance of NH4+ and carboxylic acid buffering on cloud pH when strong acid inputs decrease. It would be good to mention here that other buffers may also play an important role, including HCO3- (see, e.g., Collett et al. (1999) Internal acid buffering in San Joaquin Valley fog drops and its influence on aerosol processing. Atmos. Environ., 33, 4833-4847).

We include this now on page 9, line 30 in the revised manuscript:

"Buffering by CO2 and SO2 becomes important at pH values above 6 (Liljestrand, 1985) and buffering by higher organic acids is important in highly polluted areas (Collett et al., 1999)."

6. p. 13, lines 89-92: Again talking about the pH buffering effect, the authors point out that a factor of 2 decrease in sulfate and nitrate inputs yields a pH increase of 1 unit, with buffering, vs. 2.1 units predicted w/o buffering. Have they examined historical time series of cloud pH to see if this effect is apparent in ambient observations? One simple comparison to ballpark this might be to look at the pH trends over time in various regions of the world as summarized by Pye et al. (2020) vs. known changes in NOx and SOx emissions in those regions.

Following the referee's suggestion, we looked at the historical record at Whiteface Mountain in New York and have added the following to the discussion of the buffering effect:

"The low sensitivity of cloudwater pH to strong acidity is seen in the long-term measurements of summertime cloudwater ions at Whiteface Mountain, NY (44°22'N, 73°54'W). Between 1994 and 2013, strong acidity at the site decreased by two-thirds, from about 300 to 100  $\mu$ eq L-1, but the VWA pH increased by only 0.8 units, from 3.8 to 4.6 (Schwab et al. 2016). At the same time, [NH4+] decreased from about 125 to 60  $\mu$ eq L-1 but collocated precipitation measurements showed no trend in NH3T (Schwab et al. 2016). Carboxylate ions were not measured. These changes in cloudwater ion concentrations at Whiteface Mountain are similar to those in Fig. 5 and signify the pH buffering effect of NH4+ volatilization. Without this buffering, pH at the site would have increased by about 2.8 units."

7. The manuscript would benefit from a more detailed discussion of how cloud pH data were selected for comparison to model predictions. The authors include several observational studies in a number of regions, but certainly not all. I understand that some points are omitted due to source proximity and some, I think, were omitted due to being from an era far from the 2013 simulation and its emissions levels (although some fairly old measurements are included). Others (e.g., several studies from Japan mountain sites and the recent Kim et al. observations from the northern Pacific (https://doi.org/10.1007/s10874-020-09403-8)) are not included. A more robust discussion of why various datasets were included/excluded would give the reader greater confidence in the very promising matches shown with modeled pH values by region.

We have further clarified our criteria for selecting observations for comparison with the simulation. Page 7, lines 19-24 in the revised manuscript are as follows (underlined text added):

...measured at mountain sites, coastal sites (marine fog), and from aircraft campaigns (Table A1). We exclude continental fogwater measurements because of their sensitivity to local emissions. The measurements span the period from 1980 to 2018, but we generally exclude observations made before 2005 in East Asia, Europe and the US because of the strong emission trends in these areas. We include some older measurements in the western US and northern Europe when there are no recent measurements in the particular region and the sites are relatively remote. Some of the observations are taken from the Pye et al. (2020) compilation."

Our compilation includes observations in Japan that were collected at Mt. Tateyama in 2007–09. All other cloudwater measurements in Japan that we are aware of were made before 2005 and thus were excluded from our compilation. We thank the referee for pointing out the observations by Kim et al. (2019) in the North Pacific. We had indeed missed them in our original compilation and now include them in the revised manuscript.

**References**

Collett, J. L., Hoag, K. J., Rao, X. and Pandis, S. N.: Internal acid buffering in San Joaquin Valley fog drops and its influence on aerosol processing, Atmos. Environ., 33(29), 4833–4847, doi:10.1016/S1352-2310(99)00221-6, 1999.

Kim, H. J., Lee, T., Park, T., Park, G., Collett, J. L., Park, K., Ahn, J. Y., Ban, J., Kang, S., Kim, K., Park, S.-M., Jho, E. H. and Choi, Y.: Ship-borne observations of sea fog and rain chemistry over the North and South Pacific Ocean, J. Atmos. Chem., 76(4), 315–326, doi:10.1007/s10874-020-09403-8, 2019.

Liljestrand, H. M.: Average rainwater pH, concepts of atmospheric acidity, and buffering in open systems, Atmospheric Environ. 1967, 19(3), 487–499, doi:10.1016/0004-6981(85)90169-6, 1985.

Schwab, J. J., Casson, P., Brandt, R., Husain, L., Dutkewicz, V., Wolfe, D., Demerjian, K. L., Civerolo, K. L., Rattigan, O. V., Felton, H. D. and Dukett, J. E.: Atmospheric Chemistry Measurements at Whiteface Mountain, NY: Cloud Water Chemistry, Precipitation Chemistry, and Particulate Matter, Aerosol Air Qual. Res., 16(3), 841–854, doi:10.4209/aaqr.2015.05.0344, 2016.

Stumm, Werner., Sigg, Laura. and Schnoor, J. L.: Aquatic chemistry of acid deposition, Environ. Sci. Technol., 21(1), 8–13, doi:10.1021/es00155a001, 1987.

**Anonymous Referee #2**

Received and published: 22 July 2020

Review of "Global modeling of cloudwater acidity, rainwater acidity, and acid inputs to ecosystems" by Shah et al.

This is a modeling study aiming to present a global view of cloud and rainwater pH, with particularly important aspects of the work including improved cloudwater pH calculations, inclusion of some carboxylic acids and dust alkalinity, and an explicit rainwater pH calculation. The topic is of relevance to the journal and addresses and important issue in the field of atmospheric sciences. I found the paper to be very well written, clear, and with conclusions well supported by the analysis. I recommend publication and do not have any additional suggestions for the authors.

We thank the referee for their comments.